# Virtual Cells as Causal World Models: A Perspective on Evaluation

## Abstract

This perspective argues that evaluating AI virtual cells requires moving beyond predictive accuracy toward assessing their ability to function as causal world models of biology. Existing benchmarks emphasize fit to observed data, rewarding pattern matching but failing to test responses to interventions. We propose that trustworthy virtual cells require causal evaluation: metrics and benchmarks that assess intervention validity, counterfactual consistency, trajectory faithfulness, and mechanistic alignment. Our contribution is two-fold: (1) a survey of recent approaches to virtual cell modeling, and (2) a taxonomy of causal evaluation metrics mapped to available perturbation datasets and benchmarks. By outlining gaps and proposing unified causal benchmarks, we position causal evaluation as the key step toward making virtual cells reliable world models of biology.

## 1 Introduction

Modern biology sits at a crossroads: even with the complete genetic code and vast single-cell atlases such as the Human Cell Atlas (Regev et al., 2017), CELLxGENE (Program et al., 2025), Tahoe-100M (Zhang et al., 2025), and scPerturb (Peidli et al., 2024), our ability to predict cellular responses to drugs, mutations, or environmental change remains limited (Wen et al., 2023; Rood et al., 2024). The bottleneck is not lack of data, but models that fail to capture how biological systems actually work (Glocker et al., 2021; Listgarten, 2024). Recent biological Foundation Models (FMs) such as GeneFormer (Zheng & Gao, 2023), Evo2 (Brixi et al., 2025), scFoundation (Hao et al., 2024), and AIDO (Ellington et al., 2025) show impressive predictive power, but are often limited to a single biological layer and capture associations rather than causal mechanisms. *Despite their variety, these models remain predictive rather than causal, with evaluations centered on accuracy or likelihood rather than causal validity.* Some advanced models fail to outperform simple linear baselines (Rood et al., 2024; Peidli et al., 2024). Efforts like PerturBench (Peidli et al., 2024) have begun to standardize predictive benchmarking, but there is still no equivalent of ImageNet (Deng et al., 2009) or GLUE (Wang et al., 2018) for causal evaluation. Biology is hierarchical (i.e., genome, transcriptome, proteome, metabolome, phenome), and disregarding this interdependence yields models that are fundamentally misaligned with the underlying biology (Kitano, 2002; Hood & Flores, 2012).

This raises the motivating question: *When does a predictive model of cells become a true world model, able to answer counterfactuals and generalize beyond its training data?* Because no dataset fully captures the multilayered complexity of the cell, uncertainty is an inherent property of both biological systems and virtual models. Evaluation must therefore address not only whether a prediction is correct, but also how confident we should be in that prediction. To this end, our vision of AI virtual cells is simulation-ready representations that reason about mechanisms, predict perturbation responses, and serve as in silico testbeds (Bunne et al., 2024; Carr et al., 2024; Noutahi et al., 2025). These can be thought of as biological world models, not just reproducing observed data but answering "what if" and "how" questions.

Our contribution is two-fold: (1) a survey of recent approaches to virtual cell modeling, and (2) a taxonomy of causal evaluation metrics mapped to available perturbation datasets and benchmarks (Figure 1). We do not prescribe how to build causal virtual cells; rather, we argue that without principled evaluation, progress toward trustworthy, mechanistic virtual biology will remain directionless.

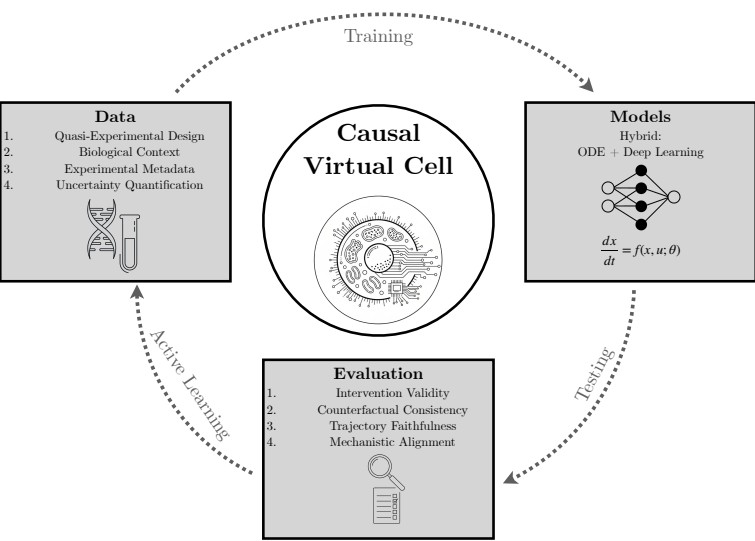

Figure 1: Summary of our proposed framework, which is described in Section 4.

## 2 RELATED WORK: PREDICTIVE APPROACHES

Predictive approaches to virtual cell modeling aim to reproduce observed cell states or transitions rather than identify or test cause–effect relationships. In this section, we review predictive models, the data used to create them, and how they are evaluated, before outlining their key limitations.

### 2.1 MODELS

Autoencoder-based and conditional architectures such as scGen (Lotfollahi et al., 2019), CPA (Lotfollahi et al., 2021), GEARS (Roohani et al., 2024), scCade (Ou et al., 2024), and scPerb (Tang et al., 2024) interpolate from control to perturbed states, while models like Biolord (Piran et al., 2024), CoupleVAE (Wu et al., 2025), SAMS-VAE (Bereket & Karaletsos, 2023), scVI (Lopez et al., 2018), and CRADLE-VAE (Baek et al., 2025b) enhance latent representations or capture combinatorial and differential perturbations. Other architectures include MichiGAN, which applies GANs for disentangled single-cell generation (Yu & Welch, 2021), CellFlow, which uses flow-matching for phenotype modeling (Klein et al., 2025), and CellOT, which applies optimal transport to map cellular trajectories (Bunne et al., 2023). Beyond autoencoders and GANs, diffusion models (Ho et al., 2020; Song et al., 2020) have been adapted for imputation, denoising, and simulation tasks.

Biological FMs build on the same generative principles but distinguish themselves by scale and scope: they are pretrained on millions of cells or sequences and fine-tuned across diverse downstream tasks, enabling broader transferability. However, these evaluations remain predictive, emphasizing reconstruction accuracy or classification performance. **Genomic/DNA** FMs are trained on DNA sequences to understand regulatory functions and predict genetic outcomes. They enable variant effect prediction, with Enformer (Avsec et al.) and Geneformer (Theodoris et al., 2023) tackling regulatory variants, and EVO2 (Brixi et al., 2025) achieving breakthrough performance on noncoding pathogenicity. FMs learn RNA sequence–structure relationships for tasks such as RNA structure/function prediction (RiNALMo (Penić et al., 2025), HydraRNA (Li et al., 2025a)), mRNA design (mRNA-FM (Li et al., 2025c)), and RNA modification site detection (AIDO.RNA (Zou et al., 2024)). **Protein** FMs predict structures, attributes, and guide design. Beyond structure, they estimate stability and binding affinity, critical for therapeutic applications. Use cases include de novo protein design (ProtGen (Madani et al., 2023), ProtGPT2 (Ferruz et al., 2022)), structure prediction (ESM-2 (Lin et al., 2023), ESM-3 (Hayes et al., 2025), MSA-transformer (Rao et al., 2021), Boltz-2 (Passaro et al., 2025), AlphaFold 3 (Abramson et al., 2024)), and single-sequence property prediction. **Single-cell** FMs analyze omics data, often across modalities, to model cellular states. Applications include cell type annotation (scBERT (Yang et al., 2022), scGPT (Cui et al., 2024), CellFM (Zeng

et al., 2025)), batch correction (scPRINT (Kalfon et al., 2025b)), and perturbation response prediction (scFoundation (Hao et al., 2024), CellFM (Zeng et al., 2025)). **Multi-modal** FMs are emerging to unify layers. SCARF integrates scRNA-seq and scATAC-seq (Liu et al., 2025), LucaOne unifies DNA, RNA, and protein (He et al., 2024), and ChatNT frames genomic tasks as text-to-text (Richard et al., 2024). Simulation-aware models such as scMultiSim (Li et al., 2025b) and Xpressor (Kalfon et al., 2025a) capture cross-modality dynamics. These highlight progress beyond single layers, but integration remains challenging, with evaluation still dominated by predictive accuracy rather than causal benchmarks.

## 2.2 DATA

The datasets highlighted here are widely used in virtual cell modeling, supporting training and evaluation of models that capture cell states or transitions without testing causal mechanisms. **Large-Scale Atlases.** Projects such as Tahoe-100M (Zhang et al., 2025) and Parse-PBMC (Parse Biosciences, 2023), now provide internally consistent datasets with millions to over one hundred million cells. The success of baseline-only efforts such as Tabula Sapiens (Quake & Consortium, 2024) and the Human Cell Atlas (Regev et al., 2017) highlights the importance of coordination across tissues and donors. Aggregation initiatives such as scBaseCount (Youngblut et al., 2025) and CELLxGENE (Program et al., 2025) have further created large harmonized resources by systematically combining hundreds of smaller public datasets. **Synthetic Data Generators.** Splatter (Zappia et al., 2017), SymSim (Zhang et al., 2019), and scDesign3 (Song et al., 2024) are increasingly used to generate controlled transcriptomic data for benchmarking predictive models. Predictive virtual cell models sometimes integrate **Clinical and Phenotypic Data**. The Cancer Genome Atlas (TCGA) (Tomczak et al., 2015), for example, has been used to link single-cell embeddings to tumor states (Tao et al., 2019; Chu et al., 2022), while UK Biobank (Bycroft et al., 2018) and EHR-derived datasets like the All of Us Research Program (All of Us Research Program Investigators, 2019) support predictive modeling of disease risk or treatment outcomes.

## 2.3 EVALUATION

Evaluation in predictive virtual cell modeling relies on established metrics and strategies that measure how well models reproduce observed cell states or transitions. We organize these into metrics that assess predictive fit (e.g., sequence modeling, classification, perturbation response) and on strategies that give these metrics meaning through baseline comparisons and generalization tests.

### 2.3.1 METRICS

Predictive virtual cell models are typically evaluated using scalar metrics that quantify how well model outputs match observed data, and provide the basic measures of predictive fit. **Sequence modeling** metrics are used by Evo 2 as a proxy for evaluating how well the predicted biological sequence distribution matches the ground truth, as we all as a measure of gene essentiality (Brixi et al., 2025). **Sequence classification** metrics are computed by RNA FMs for distinguishing the introns vs. exons regions, splice variations, and splice variations (Chen et al., 2022a). Categorical cross entropy, for instance, can be used to assess the predicted output distribution in relation to labels from the set of class labels. **Epigenome prediction** is a common task which requires predicting expression values to compare to the ground truth expression values (Lotfollahi et al., 2019; 2021; Roohani et al., 2024; Ou et al., 2024; Tang et al., 2024). Mean absolute error, mean squared error (MSE), $R^2$, and cosine similarity are commonly used as metrics for regressing continuous expression values. Detection metrics are applied to the prediction of genetic interaction (Roohani et al., 2024). **Subcellular localization** evaluates predictions of spatial cell properties by comparing a set of predicted, labeled 2D Euclidean clusters to the ground-truth labeled cellular subcomponets. Adjusted rand index (ARI) and adjusted mutual information are used to evaluate the SubCell (Gupta et al., 2024), and average probability of correct label is used to evaluate DeepProfiler (Tomkinson et al., 2024) and CellProfiler (Stirling et al., 2021). **Macroscopic cell state detection** (of cell type and cell health, for example) is also commonly used as a benchmark for virtual cell models (Brixi et al., 2025). This involves comparing the predicted per-state logistic labels, to the binary ground truth labels. Typical retrieval metrics are employed by (Brixi et al., 2025), such as recall, precision, F1, ROC-AUC, etc. Similarly, drug Mechanism-of-Action (MoA) is also used as detection benchmark in the same fashion. **Epigenome delta prediction** evaluates models on their ability to

predict perturbation outcomes. Here, epigenetic deltas are extracted from the model and compared against the ground truth deltas. Fold-change, log fold-change, DE overlap accuracy, directionality agreement, Wilcoxon rank-sum, and Top-$k$ precision are commonly applied metrics in this setting (Adduri et al., 2025; Ou et al., 2024; Tang et al., 2024).

### 2.3.2 STRATEGIES

Evaluation strategies define how scalar metrics are applied to assess model capability and generalization. Metrics provide raw measures of predictive fit, while strategies organize them into benchmarks, baseline comparisons, and ablations that guide model selection and assess genuine progress. **Rank based metrics.** As noted by PerturBench (Wu et al., 2024), scalar metrics on epigenome prediction often wash out signal and may encourage effects like "mode collapse." Rank-based interpretations (e.g., Log-FC, cosine similarity) better capture differences and align with a common use of virtual cell models: ranking perturbations by effect size.

**Calibration.** Many virtual cell models (e.g., scGen (Lotfollahi et al., 2019), CPA (Lotfollahi et al., 2021), GEARS (Roohani et al., 2024)) are probabilistic, making calibration crucial. Measuring calibration helps weight predictions by uncertainty and build trust. Negative log-likelihood can be used for sequence metrics, while Expected Calibration Error (ECE) applies to classification tasks (Naeini et al., 2015).

### 2.4 LIMITATIONS OF PREDICTIVE APPROACHES

Predictive frameworks excel at interpolating and extrapolating trajectories but remain black boxes that lack mechanistic explanations (Moran & Aragam, 2025). They perform well on held-out data yet struggle to generalize to unseen perturbations or conditions (Jiao et al., 2024; Tejada-Lapuerta et al., 2025) and they predict outcomes without testing causal guarantees or answering counterfactual questions (Laubach et al., 2021). These limits reflect the data: most resources are observational or transcriptome-only with few true interventions (Rawal et al., 2025); multi-omic and temporal datasets remain scarce (Carr et al., 2024); and scRNA-seq yields only static snapshots, preventing before–after comparisons (Noutahi et al., 2025). Most datasets capture a single layer, leaving genome-to-proteome mechanisms unevaluated, while the combinatorial complexity of perturbations demands coordinated community efforts (Tejada-Lapuerta et al., 2025).

Evaluation is likewise dominated by predictive metrics such as MSE, $R^2$, and log-likelihood, which capture correlations but not mechanisms (Goshisht, 2024). Even perturbation benchmarks emphasize regression measures, insufficient for mechanistic alignment (Noutahi et al., 2025). Newer metrics, uncertainty quantification (e.g., calibration error (Yao et al., 2019)), distributional similarity (e.g., MMD (Gretton et al., 2012)), and rank-based evaluation (e.g., LogFC rank in PerturBench (Peidli et al., 2024)) are progress but still treat predictions as point estimates. In practice, uncertainty guides how results are used: low-confidence predictions signal the need for more data or refinement, while high-confidence results provide greater justification for moving forward. Uncertainty is therefore a cross-cutting dimension of evaluation, shaping how validity, consistency, and mechanistic alignment are interpreted.

## 3 CAUSAL METHODS

Compared to predictive methods that reproduce observed patterns, causal models aim to capture cause–effect relationships and are judged on whether they reproduce intervention outcomes or generate counterfactuals consistent with known mechanisms (Zanga et al., 2022; Carr et al., 2024; Pearl, 2012; Bareinboim & Pearl, 2016; Niu et al., 2024). See Figure 2 for a visual comparison of predictive and causal approaches. Causal machine learning offers a path forward by treating perturbations as structured interventions and seeking mechanisms invariant across environments (Glymour et al., 2019; Tejada-Lapuerta et al., 2025). Causality in biology can be defined in complementary ways. (i) A *mechanistic* view emphasizes biochemical interactions and dynamical processes (e.g., MAPK phosphorylation cascades that link receptor activation to downstream gene expression) (Tejada-Lapuerta et al., 2025). (ii) A *probabilistic* view emphasizes conditional independences in observational data (e.g., ERK activation being independent of receptor status once Ras activity is accounted for) (Glymour et al., 2019). (iii) A *counterfactual* view highlights potential outcomes under

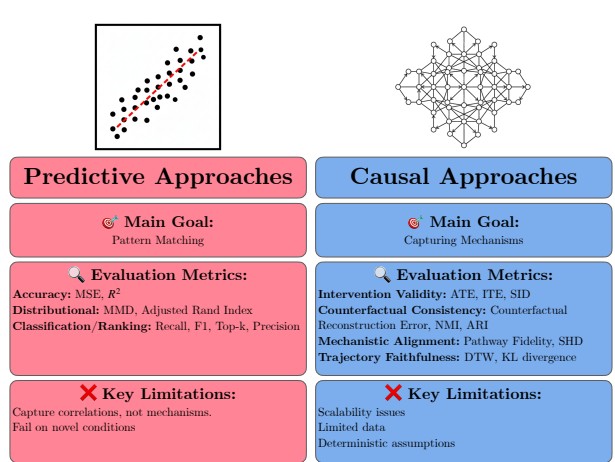

Figure 2: Comparison of predictive (Section 2) and causal (Section 3) approaches.

interventions (e.g., asking how a tumor cell's transcriptome would change if KRAS were knocked out versus left intact) (Lobentanzer et al., 2024).

## 3.1 CAUSAL MODELS

Structural Causal Models (SCMs) represent variables as directed graphs with explicit rules for interventions via the do-operator (Pearl, 2012; Rawal et al., 2025). Dynamical causal models extend this framework, using ordinary or stochastic differential equations to describe how biological states evolve under perturbation (Tejada-Lapuerta et al., 2025). Together, these perspectives form the foundation for causal virtual cells: models that not only predict cellular responses but also explain them in terms of mechanisms that remain invariant across conditions.

We highlight four broad families of causal models relevant to virtual cells. Early causal models in systems biology and pharmacology used **ODE-Based Models.** Early causal models used ODEs to describe biochemical networks (Kitano, 2002; Alon, 2019), with classical examples in electro-physiology and metabolism (Hodgkin & Huxley, 1952; Strassberg & DeFelice, 1993; Noble, 1960; Courtemanche et al., 1998; Higgins, 1964; Heinrich & Rapoport, 1974; Adamczyk et al., 2011). Recent methods adapt to single-cell gene regulatory networks (GRN) inference and trajectories: SCODE (Matsumoto et al., 2017), GRISLI (Aubin-Frankowski & Vert, 2020), SINCERITIES (Papili Gao et al., 2018); RNA-velocity extensions include scVelo (Bergen et al., 2020), UniTVelo (Gao et al., 2022), Velorama (Singh et al., 2024), DynaMO (Kuang et al., 2018). GraphDynamo (Zhang et al., 2023) and STORM (Peng et al., 2024) add graph/stochastic structure. SDEs help with noise but raise scalability/identifiability challenges (Komorowski et al., 2011; Browning et al., 2020; Persson et al., 2022). **Hybrid Causal Deep Learning Models** address the scalability limits of mechanistic models by integrating neural networks. Neural ODEs (Chen et al., 2018) flexibly parameterize dynamics, Latent ODEs (Rubanova et al., 2019) extend this to hidden states, and Universal Differential Equations (UDEs) (Rackauckas et al., 2020) embed neural nets within ODEs to learn unknown processes while preserving structure. In single-cell biology, DeepVelo (Chen et al., 2022b) extends RNA velocity with neural ODEs, PerturbODE (Lin et al., 2025) models perturbation dynamics, and PHOENIX (Hossain et al., 2024) integrates lineage information. Knowledge-primed neural networks, including sparse MLPs and graph-informed architectures, constrain learning with pathway priors (Fortelny & Bock, 2020). **Graphical and Counterfactual approaches** represent cellular dependencies as Directed Acyclic Graphs (DAGs) or SCMs. Causal discovery methods aim to recover such graphs: constraint-based algorithms such as Peter-Clark and Fast Causal Inference (Spirtes & Glymour, 1991; Spirtes et al., 2000), score-based approaches like GES (Chickering, 2002), and differentiable DAG learners including NOTEARS (Zheng et al., 2018), DAG-GNN (Yu et al., 2019), and GraN-DAG (Lachapelle et al., 2019). Applications to single-cell data remain early but are grow-

ing: CausalCell (Wen et al., 2023) integrates multiple strategies for GRN inference, LINEAGEOT (Forrow & Schiebinger, 2021) combines lineage tracing with optimal transport, and CARDAMOM (Yuan & Duren, 2025) applies a Bayesian SCM-inspired framework. Invariance-based methods, including ICP (Peters et al., 2016), Causal Dantzig (Rothenhäusler et al., 2019), and anchor regression (Rothenhäusler et al., 2021), identify gene modules stable across environments. **Causal Perturbation Prediction models** embed causal structure into predictive architectures, enabling counterfactual simulation and enforcing invariance. scCausalVI (An et al., 2025) disentangles baseline heterogeneity from treatment effects using a variational inference framework guided by SCM principles, while CausCell (Gao et al., 2025) combines SCMs with diffusion-based generative modeling to generate counterfactual single-cell states. CINEMA-OT (Dong et al., 2023) leverages independent component analysis and optimal transport to separate confounding from treatment effects. Other methods extend this paradigm: GPO-VAE (Baek et al., 2025a) aligns VAE latent spaces with GRN priors, GraphVCI (Wu et al., 2022) predicts counterfactual responses on graphs, and DCD-FG (Lopez et al., 2022) infers factor graphs with causal constraints.

## 3.2 CAUSAL DATA

Unlike predictive resources in Section 2.2, which are mostly observational or baseline-only, causal modeling requires datasets with explicit interventions, perturbations, or synthetic counterfactuals. These form the basis for testing whether models capture cause–effect relationships rather than correlations. Perturbation datasets provide the closest analogue to randomized controlled trials in cell biology (Laubach et al., 2021). High-throughput CRISPR-based screens such as Perturb-seq (Dixit et al., 2016; Adamson et al., 2016) and its large-scale extensions (Replogle et al., 2022), as well as Optical Pooled Screens (OPS) (Feldman et al., 2019), have become cornerstones of interventional single-cell data. Recent large-scale initiatives such as X-Atlas (Huang et al., 2025) extend Perturb-seq to tens of millions of cells, providing a reference-scale atlas of genetic perturbations that could serve as a benchmark for causal modeling. These datasets enable direct measurement of how cellular systems respond to interventions, though they remain sparse, noisy, and limited to subsets of possible perturbations. For causal inference, single-modality measurements (e.g., transcriptomes) are often insufficient, as mechanisms span multiple regulatory layers. Emerging perturbational datasets incorporate multi-omic readouts, including joint measurements of RNA and protein (perturbational CITE-seq (Stoeckius et al., 2017; Hao et al., 2021)), and chromatin accessibility (Perturb-ATAC (Rubin et al., 2019)).

## 3.3 EVALUATION

Evaluation of causal virtual cells requires metrics and strategies that assess whether models capture underlying mechanisms, respect known biological pathways, and generalize to unseen interventions.

### 3.3.1 METRICS

**Intervention Validity** measures whether the model reproduces observed outcomes under experimental interventions (e.g., CRISPR knockouts, drug perturbations). This can be tested through effect size and causal effect estimation, including (Individual, Average, Conditional Average) Treatment Effects (ITE, ATE, CATE; respectively), and population-level summaries such as log Fold-Change (LogFC) (Hill, 2011; Shalit et al., 2017; Winship & Morgan, 1999; Hernán & Robins, 2006). Attribution accuracy and regression coefficients further quantify whether responses are correctly attributed to latent or confounding factors (Johansson et al., 2016; Louizos et al., 2017; Schölkopf et al., 2021). Distributional alignment metrics complement these by comparing predicted and observed interventional outcomes: Structural Intervention Distance (SID, which measures discrepancies in interventional distributions (Sachs et al., 2005; Peters & Bühlmann, 2013)) (Hauser & Bühlmann, 2012), Maximum Mean Discrepancy (MMD) (Gretton et al., 2012), energy distance (Székely & Rizzo, 2013), and cluster-preservation indices (e.g., ARI (Hubert & Arabie, 1985)) assess how well causal structure and response space are preserved. **Counterfactual Consistency** quantifies whether counterfactual predictions are biologically plausible, mechanistically grounded, and consistent with both simulated and experimental causal effects. Evaluation involves: (i) *counterfactual reconstruction error*, which compares predicted states against observed perturbation responses, using metrics such as Pearson correlation, MSE, Normalized Mutual Information (NMI), ARI, and marker gene preservation (Gayoso et al., 2022; Lotfollahi et al., 2019); (ii) *latent disentan-*

*glement scores*, to assess how causal factors are separated in latent space, and are quantified via clustering and silhouette-based indices (Bengio et al., 2019; Gao et al., 2025; An et al., 2025); and (iii) *agreement with ground-truth*, benchmarked against (semi)synthetic datasets such as GeneNetWeaver (Schaffter et al., 2011), SynTReN (Van den Bulcke et al., 2006), PerturBench (Wu et al., 2024), and real-world intervention datasets (e.g., Sachs flow cytometry, Perturb-seq). **Trajectory Faithfulness** measures alignment between predicted and observed time-resolved responses. Perturb-seq, OPS, and synthetic benchmarks such as DREAM4 (Greenfield et al., 2010) and SynTReN (Van den Bulcke et al., 2006) provide the experimental and simulated foundations for evaluating trajectory faithfulness. Evaluation metrics include: (i) *trajectory similarity*, using Dynamic Time Warping (DTW), KL divergence of state distributions, and optimal transport to compare predicted versus experimental temporal profiles (Cuturi, 2013; Chen et al., 2018); (ii) *trend alignment*, where Pearson correlation, MSE, and RMSE quantify concordance between predicted and observed expression dynamics, including treatment effects (Lotfollahi et al., 2019; An et al., 2025); and (iii) *structural consistency*, such as SID and causal graph recovery scores assess whether perturbation trajectories follow known pathways (Peters et al., 2016). **Mechanistic Alignment** quantifies overlap between known pathways and mechanistic constraints. Evaluation includes: (i) *pathway fidelity scores*, measure overlap between inferred interactions and curated databases such as KEGG (Kanehisa, 2002) and Reactome (Fabregat et al., 2018), and assess whether models recover literature-supported mechanisms; (ii) *invariance tests*, evaluate the stability of causal predictions across perturbations, using conditional independence checks, out-of-distribution generalization, and robustness to modality or context shifts (Peters et al., 2016; Heinze-Deml et al., 2018); and (iii) *causal graph similarity*, using metrics like SID and Structural Hamming Distance (SHD). **GRN Recovery** tests whether models recover both the statistical associations and causal intervention structure underlying biological regulatory graphs like GRNs. Standard measures include: (i) *edge prediction accuracy*, with AUROC and AUPR quantifying discrimination between true and false regulatory edges across thresholds; (ii) *graph distance metrics*, such as SHD and SID; and (iii) *benchmark datasets*, including DREAM4 challenges (Greenfield et al., 2010) and GeneNetWeaver (Schaffter et al., 2011) simulations, which provide community standards for comparing GRN inference methods.

### 3.3.2 STRATEGIES

Causal evaluation strategies define how metrics are applied to probe causal validity. They specify the setups, tasks, and comparisons that reveal whether models generalize beyond observed data. **Synthetic ground-truth tests** enable precise quantification of GRN recovery and counterfactual consistency. Simulation frameworks such as GeneNetWeaver (Schaffter et al., 2011), SERGIO (Dibaeinia & Sinha, 2020), and scDesign3 (Song et al., 2024) generate datasets with known causal graphs. **Pathway fidelity tasks** evaluate whether models preserve mechanistic structure by testing predicted perturbation effects against curated pathways (e.g., KEGG (Kanehisa, 2002), Reactome (Fabregat et al., 2018), BioModels (Le Novere et al., 2006)). **Invariance-based evaluation** tests whether predictions remain stable across environments or cell contexts, using causal discovery frameworks such as ICP (Peters et al., 2016) and anchor regression (Rothenhäusler et al., 2021). **Generalization regimes** are examined under unseen single perturbations, novel combinations, and temporal holdouts. These tasks parallel predictive benchmarks but require causal consistency rather than fit alone (Arjovsky et al., 2019; Lotfollahi et al., 2019; Schölkopf et al., 2021; An et al., 2025). For **baselines and ablations**, causal models are compared against predictive-only baselines (e.g., sc-Gen (Lotfollahi et al., 2019), CPA (Lotfollahi et al., 2023)), to test whether causal inductive biases improve counterfactual validity. Component ablations (e.g., removing causal regularizers, pathway priors) clarify which features drive causal performance (An et al., 2025; Gao et al., 2025). **Perturbation benchmarks** enable systematic evaluation of interventional datasets (e.g., Perturb-seq, OPS). Frameworks such as PerturBench (Wu et al., 2024) and OP3 (Szałata et al., 2024) provide standardized tasks for perturbation response prediction, with OP3 emphasizing causal evaluation criteria such as intervention validity and counterfactual prediction. **General causal benchmarks** include broader efforts such as CausalBench (Wang, 2024), which provide reference standards for evaluating causal inference methods across domains, including perturbation modeling.

### 3.4 CURRENT LIMITATIONS OF CAUSAL APPROACHES

Causal models for virtual cells provide interpretability and mechanistic grounding but remain limited by strong assumptions and scalability issues (Bunne et al., 2024; Carr et al., 2024; Lan et al.,

2025; Noutahi et al., 2025). Many ODE-based and hybrid methods assume acyclicity or causal sufficiency (Michoel & Zhang, 2023; Wen et al., 2023; Tejada-Lapuerta et al., 2025), restricting feedback loops and hidden confounders. They also rely on idealized interventions and face unresolved parameter identifiability challenges (Klipp & Liebermeister, 2006). Consequently, most approaches remain confined to small circuits, velocity-style embeddings, or low-dimensional summaries rather than genome-wide, multi-omic contexts (Glymour et al., 2019; Lobentanzer et al., 2024; Lan et al., 2025). Causal data availability remains a bottleneck (Carr et al., 2024). Perturbation assays such as Perturb-seq and OPS expand access to interventional data but are sparse, noisy, and context-biased. Ground-truth causal graphs are rare, temporal measurements limited, and destructive assays like scRNA-seq prevent before–after comparisons. Synthetic benchmarks help but cannot fully capture biological complexity or generalize to real systems (Cheng et al., 2022). Evaluation remains fragmented: efforts emphasize GRN recovery, pathway fidelity, or counterfactual validation, but no unified taxonomy of causal metrics exists for virtual cells (Bunne et al., 2024). Most evaluations also treat outcomes as deterministic, even though biological systems and perturbational data are inherently uncertain. Noisy interventions, incomplete priors, and hidden confounders require models and metrics to propagate uncertainty; otherwise, causal models risk overstating confidence in fragile or context-specific findings. Overall, causal approaches remain proof-of-concept; without standardized datasets, metrics, and benchmarks, virtual cells cannot yet reliably test mechanisms over correlations (Rawal et al., 2025).

## 4 PROPOSED FRAMEWORK

### 4.1 MECHANISTIC APPROACHES TO MODEL DESIGN

The ambition for virtual cells is to represent cellular machinery in mechanistic detail, ideally as systems of differential equations capturing causal interactions and dynamics (Klipp et al., 2005; Alon, 2019). ODEs assume deterministic dynamics and face the "curse of dimensionality," making whole-cell simulation infeasible (Waltemath et al., 2011; Tomita et al., 1999). Extensions to SDEs capture intrinsic noise and uncertainty, essential for models that must quantify confidence as well as mean behavior. Progress will require hybrids that combine mechanistic grounding with deep learning flexibility. Universal and neural ODEs (Rackauckas et al., 2020; Chen et al., 2018) integrate biological priors with neural architectures, while causal constraints, sparsity, and disentangled representations improve interpretability (Brunton et al., 2016; An et al., 2025). Crucially, model design is inseparable from evaluation: benchmarks must test not only predictive accuracy but also causal validity (Peters et al., 2017; Schölkopf et al., 2021), ideally within a lab-in-the-loop paradigm where models are iteratively refined with experiments (Frey et al., 2025; Chandak et al., 2023).

### 4.2 CAUSAL EVALUATION FROM ESTABLISHED DATA

In an ideal setting, causal evaluation would use multi-omic interventional time-series data with matched controls and rich context. A fundamental challenge is that most widely available datasets are observational, whereas causal inference requires interventional data (Rawal et al., 2025). Below, we propose four improvements to leverage existing data.

**Quasi-Experimental Design** can strengthen existing observational resources with matched controls to approximate causal contrasts. Propensity score matching (Rosenbaum & Rubin, 1983), paired sampling (Rubin, 1974), and distributional methods like optimal transport (Peyré et al., 2019) (exemplified by CINEMA-OT (Dong et al., 2023)) illustrate how confounders can be separated from perturbation effects to reconstruct counterfactual states. The goal is not full causal identification, but extending robust statistical tools to high-dimensional single-cell settings. Furthermore, most assays capture only static snapshots, so obtaining temporal anchors and allowing evaluating trajectory faithfulness requires proxies such as pseudotime (Trapnell et al., 2014; Saelens et al., 2019), RNA velocity (La Manno et al., 2018; Bergen et al., 2020), dose–response designs (Subramanian et al., 2017), and repeated sampling.

**Biological Context Enhancement** (in the absence of large-scale multi-omic interventional datasets) can capture interdependencies across molecular layers. The following strategies offer partial solutions: (i) *Structured priors*, such as KEGG (Kanehisa, 2002), Reactome (Fabregat et al., 2018), STRING (Szklarczyk et al., 2021), and BioGRID (Oughtred et al., 2019), which provide pathway

and interaction knowledge for fidelity tests. Meanwhile, ontologies such as GO (Consortium, 2019) and Cell Ontology (Diehl et al., 2016)) enable dataset alignment, and domain-specific LMs like BioBERT (Lee et al., 2020) enrich metadata. (ii) *Synthetic data-based* tools such as GeneNetWeaver and DREAM (Schaffter et al., 2011; Marbach et al., 2012), SERGIO (Dibaeinia & Sinha, 2020), DYNGEN (Cannoodt et al., 2021), and scDesign3 (Song et al., 2024) simulate perturbations and multi-omic readouts, providing ground truth for benchmarking.

**Experimental Metadata** helps discriminate between experimental variation and true biological signal. Examples of models that explicitly take these variations into account can be found in (An et al., 2025), (Gao et al., 2025), (Korsunsky et al., 2019), (Hao et al., 2021), and (Lopez et al., 2018). The following strategies help prepare datasets to provide this context: (i) *Metadata integration* on batch effects, protocols, and sample handling (GEO (Edgar et al., 2002), ArrayExpress (Parkinson et al., 2009), CELLxGENE (Program et al., 2025)) can stratify analyses; protocol-aware covariates improve comparability across assays (e.g. 10x vs. Smart-seq2) (Hicks et al., 2018). (ii) *Quality control and robustness*, such as UMIs, features, mitochondrial fraction, improve reliability (Luecken & Theis, 2019), and invariance-based methods such as ICP (Peters et al., 2016) and anchor regression (Rothenhäusler et al., 2021) test whether relationships remain stable across conditions.

**Uncertainty Quantification** (UQ) is essential to distinguish true signals from noise. While UQ *alone* does not yield causal models, it improves robustness in data-sparse regimes and guides experiment design. Approaches include: (i) Bayesian inference, (ii) Gaussian processes (iii) Ensembles and resampling (iv) Calibration (v) Information-theoretic scores (e.g. entropy, mutual information (BALD), and sensitivity indices (Houlsby et al., 2011)) (vi) Simulation-based inference (SBI) likelihood-free methods (Cranmer et al., 2020) quantify uncertainty in complex mechanistic models, with applications to stochastic gene expression (Toni et al., 2009), signaling dynamics (Golightly & Wilkinson, 2005), and single-cell electrophysiology (Lueckmann et al., 2017). Together, these methods enable virtual cells to attach explicit confidence to hypotheses, prioritize robust discoveries, and guide experimental validation in a lab-in-the-loop paradigm.

### 4.3 UNCERTAINTY-AWARE CAUSAL EVALUATION

A critical step is to adapt existing metrics to be uncertainty-aware, bridging current practice with the needs of causal virtual cells. For **intervention validity**, measures such as effect size correlation, treatment effect error, or distributional distances (Hill, 2011) could be extended with calibration (e.g., ECE (Naeini et al., 2015), Brier score (Glenn et al., 1950)), variance-aware distances, or likelihood-based comparisons of full distributions. For **counterfactual consistency**, where outcomes are unobservable, models should indicate high uncertainty for far out-of-distribution queries rather than overconfident predictions. For **trajectory faithfulness**, metrics such as DTW (Berndt & Clifford, 1994) or KL divergence (Kullback & Leibler, 1951) assume precise trajectories, but destructive assays prevent true before/after comparisons; evaluation should propagate error over time and flag uncertain regions in dose–response or developmental dynamics. For **mechanistic alignment**, pathway fidelity scores and graph distances like SHD and SID are deterministic; uncertainty-aware versions would weight edges by confidence, assigning higher certainty to well-established interactions (KEGG (Kanehisa, 2002), Reactome (Fabregat et al., 2018)) and lower to novel ones.

## 5 DISCUSSION & CONCLUSION

Causal evaluation is the critical test of whether AI virtual cells can evolve from predictive simulators into trustworthy world models of biology. We outlined a taxonomy of causal metrics, emphasizing uncertainty as a cross-cutting principle. Standardized benchmarks that integrate interventions, trajectories, multi-omic context, and uncertainty are essential for robustness, interpretability, and translational impact. Without them, virtual cells remain unproven; with them, they can become reliable engines for discovery and therapeutic innovation. Embedding uncertainty at every level ensures evaluation asks not only 'was the prediction correct?' but also 'how certain should we be, and what should we do next?', providing the foundation for virtual cells that are not just predictive, but trustworthy and actionable.

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
