# OpenReview forum: "VIRTUAL CELLS AS CAUSAL WORLD MODELS: A PERSPECTIVE ON EVALUATION"
_ICLR.cc/2026/Conference — ICLR 2026 Conference Withdrawn Submission_

### Official Review · Reviewer_tm4f · 2025-10-30

**Soundness:** 3
**Presentation:** 1
**Contribution:** 3
**Rating:** 2
**Confidence:** 3

**Summary:**

The authors focus on virtual cell modeling, a field of research that builds models to simulate some aspects of biological cells, generally aiming to produce realistic cell states and transitions.  While this field is growing, evaluation of virtual cells has remained predictive, focusing on correlation rather than causation.  For a virtual cell model to truly represent how cells behave and interact, it needs to represent how components causally affect each other.  To promote this shift in focus for the field, the authors first conduct a survey of current practice in virtual cell modeling, breaking down current practice in terms of modeling approaches, data, and evaluation metrics.  They then also break down current practice in causal modeling by modeling approaches, data, and evaluation metrics and discuss how virtual cell modeling evaluation can be expanded to capture causality.

**Strengths:**

The breadth of the presented survey is impressive.  The authors did a great job of covering a range of virtual cell modeling methods, evaluation metrics, and causal modeling methods, allowing this paper to serve as a rich source of references.

For the most part, the writing quality is high, making it an easy read.

**Weaknesses:**

**Poor organization as a survey paper**

While I think the authors have a strong handle on the literature in virtual cell modeling and causal modeling, and the pieces are mostly here for a solid survey paper, the way it is presented prevents the paper from really functioning as a useful survey.  Ideally, after reading Section 2, I should have a decent understanding **(1)** The problem definition(s) of virtual cell modeling, **(2)** the broad classes of approaches that people have applied to virtual cell modeling, **(3)** what sort of predictions these approaches are being used to generate, **(4)** what types of data each of these modeling methods uses/is evaluated on, and **(5)** what metrics are used to evaluate each of these model/prediction types.  However, breaking these down:

**(1) The problem definition(s) of virtual cell modeling:** This doesn't seem to be covered at all.  The only definition I can find of what the scope of "virtual cell modeling" is in the second paragraph of the introduction: "our vision of AI virtual cells is simulation-ready representations that reason about mechanisms, predict perturbation responses, and serve as in silico testbeds".  However, this seems to be the authors ideal vision of what virtual cells should be, rather than how they are actually defined in the literature.  From reading the paper, they seem to broadly be any model that focuses on making predictions about cell state and behavior.  However, if that's the definition the authors are working with, that encompasses a wide range of possible task definitions, which doesn't really seem to be discussed anywhere.

**(2)-(4) existing virtual cell modeling approaches, what predictions they produce, and what type of data they require:** are covered in part by Section 2.1.  However, while 2.1 does a great job at provided a large number of references, there's not enough description or categorization for a survey paper.  2.1 seems to mostly group models into either "Autoencoder-based and conditional architectures" (paragraph 1) and "Biological FMs" (paragraph 2), with a few bold headers in paragraph 2 separating types of FMs.  However, the vast majority of methods listed have very little information provided about them, making it very hard to parse for someone not particularly familiar with virtual cell modeling.  The bold headings do break model categories down by what type of data they are trained on and provide a few words each describing what task they are performing/what they are modeling, which is useful context.  However, the presentation method of a single dense paragraph makes the relevant information harder to extract.

**(4) What types of data each of these modeling methods uses/is evaluated on:** is also partially covered by Section 2.2.  However, it seems unlikely that all of the datasets described in 2.2 are applicable to all the models/applications discussed in 2.1.  If a similar categorization scheme were used in both 2.1 and 2.2, it would be easier to map datasets to relevant modeling approaches.

**(5) what metrics are used to evaluate each of these model/prediction types:** encounters similar difficulties to (4): while there are many categories of evaluation metrics discussed in 2.3.1, the categories here are again different from those presented in 2.1, making it hard to map evaluation metrics to modeling approaches.  The categories in 2.3.1 (broken down by task type) actually seem like great categories that could be used in 2.1 as well.

I think Section 2 really needs a table, the kind that is present in many survey papers.  There are many ways to do, but something that defines clear categories of modeling methods and maps those to the type data they use, predictions they can produce, evaluation metrics typically used, etc.  If the categories are in practice too blurred to make a clear distinction like that, or if, counter to how 2.1 seems, all of the modeling methods are flexible enough to run with any of the dataset types and evaluation metrics discussed, then that should be discussed and explicitly stated.

The paper as-is seems to assume that the reader is already familiar with the virtual cell modeling domain.  If that is the intended audience, and this lack of clarity isn't felt by the other reviewers, then you can discount these comments.  However, if you're aiming for a broader audience, then the descriptions of modeling methods 2.1 aren't clear enough.  For example, in the first sentence of 2.1, you list a group of methods that "interpolate from control to perturbed states" and another group that "enhance latent representations or capture combinatorial and differential perturbations."  Both of these descriptions are far too vague for me to understand what they're referring to.  Does "interpolate from control to perturbed states" mean that they take input data that includes example interventions (i.e., control) and learn to estimate what the resulting perturbed states would be?  "Enhance latent representations" of what?  Then in Section 2.3.1, there are sentences like "Detection metrics are applied to the prediction of genetic interaction", without actually saying what a "detection metric" is.

---
**Unclear second contribution**

The authors second contribution, in the 3rd paragraph of the introduction, is listed as "a taxonomy of causal evaluation metrics mapped to available perturbation datasets and benchmarks (Figure 1).  However, Figure 1 doesn't seem to be taxonomy of evaluation metrics.  It is instead labeled as a "Summary of our proposed framework as described in Section 4".  However, I also don't see a clear framework in Section 4.  Section 4 instead seems to consist of (1) an explanation that causality is important, (2) a list of 4 ways that virtual cell modeling can move in a causal direction, and (3) an explanation of the importance of representing uncertainty.  These are all fine points, but none of them seem anywhere close to a new "framework".

It looks like the proposed taxonomy may be the terms listed under "evaluation" in Figure 1?  These do correspond to the first 4 bold headings in Section 3.3.1, which is helpful.  However, I'm then not sure of "GRN Recovery" should also be in Figure 1.  Also, the 2nd contribution also mentions that the provided taxonomy is "mapped to available perturbation datasets and benchmarks".  However, the categorization used in 3.3.1 (and seen in Figure 1) isn't actually presented in the text of the paper until 3.3.1, and the perturbation datasets are discussed in Section 3.2, so, unless I'm missing something, the mapping between the taxonomy and the datasets is never made explicit.

Section 4 falls victim to a similar issue as Section 2 (presenting a wide range of references in dense paragraphs, with models, data, and metrics presented with different categorizations, making it hard to link them together), but it was less of an issue for me here since I'm more familiar with that literature.  Still, it would benefit from a similar reorganization as Section 2, ideally with some sort of table.

---

**Miscellaneous issues**

One of the arrows in Figure 1 is "Active Learning", a term that doesn't appear anywhere else in the paper that I can tell.

In the conclusion, the authors state that they "emphasize uncertainty as a cross-cutting principle".  However, uncertainty isn't really discussed until the very end of the paper (the final paragraph in Section 4.2 and Section 4.3), so it really doesn't come across as a "cross-cutting principle".  In addition, those parts of the paper that do discuss uncertainty feel very disjointed.  The "Uncertainty Quantification" section reads mostly like a list of techniques/approaches that deal with uncertainty/confidence in various ways (e.g., "Calibration", "Gaussian processes"), but without any discussion of how they can be applied to causal modeling or virtual cells.  Only "simulation-based inference" actually has any real discussion.  Section 4.3 is a bit more specific, but even still, it reads mostly as a future work section, suggesting that certain evaluation metrics could be extended in different ways.  These are interesting points, but their hypothetical nature means that they really don't serve as a "cross-cutting principle".

The discussion of "strategies" in Section 2.3.2 feels odd.  While the concept of evaluation strategies as the authors describe them seems sound, all that's discussed in this section is "rank based metrics" (which seems like a type of metric, not a strategy for applying a metric) and "calibrations", which seems like something that should be done for a model but again, not really a "strategy" (at least, not as the authors seem to define a strategy).

---

Ultimately, I think the pieces are here for a solid paper.  However, the current presentation is significantly holding it back.  The key insights of this paper (what the current practice is in virtual cell modeling, what alternate approaches are available in the causal modeling world, and what concrete actions can be taken to move virtual cell evaluation in a more causal direction) are buried in dense reference-heavy and description-light paragraphs, making them hard to glean without spending significant effort and limiting the likely impact of this work.

**Questions:**

No questions

---

### Official Review · Reviewer_Yd6o · 2025-11-01

**Soundness:** 2
**Presentation:** 1
**Contribution:** 2
**Rating:** 6
**Confidence:** 3

**Summary:**

This is a perspective/position paper arguing that evaluation of “AI virtual cells” must move beyond predictive fit to explicit causal assessment. The authors (i) survey recent predictive and causal approaches for virtual-cell modeling and (ii) propose a taxonomy and framework for causal evaluation built around four axes: Intervention validity, Counterfactual consistency, Trajectory faithfulness, and Mechanistic alignment, with uncertainty treated as a cross‑cutting principle.

**Strengths:**

1. Timely and well‑motivated problem framing. The distinction between predictive fit and mechanistic/causal validity is crisply articulated, with clear failure modes of purely predictive assessments (e.g., generalization to unseen interventions).

2. Actionable taxonomy of evaluation axes. The four axes and their associated metrics/strategies are specific enough to guide practitioners toward more probing tests (e.g., using SID/SHD, pathway fidelity, invariance‑based tests), not just MSE/LogFC.

3. Uncertainty as a first‑class concern. Treating calibration and distributional uncertainty as integral to causal evaluation (not an afterthought) is a valuable emphasis.

**Weaknesses:**

1. No instantiated benchmark or code. The paper proposes a taxonomy and mentions candidate datasets/benchmarks (Perturb‑seq, OP3, PerturBench), but does not release a concrete evaluation suite (tasks/splits/metrics scripts) or re‑evaluate representative models within the proposed framework. This limits immediate impact and testability.

2. Lack of empirical case studies. There is no demonstration that the proposed metrics change conclusions relative to standard predictive metrics (e.g., a re‑ranking of methods by intervention validity or SID on a public perturbational dataset).

3. Ambiguity at scale. Practical guidance for computing graph‑level metrics (e.g., SID/SHD) and pathway‑fidelity at genome scale with noisy annotations is limited; identifiability and confounding are acknowledged but not operationalized into robust scoring protocols.

**Questions:**

see above

---

### Official Review · Reviewer_EzGt · 2025-11-04

**Soundness:** 3
**Presentation:** 2
**Contribution:** 2
**Rating:** 6
**Confidence:** 4

**Summary:**

This perspective paper advocates for better evaluation of AI "virtual cells". The authors focus on causal evaluation of virtual cells, where these models are evaluated according to their ability to predict responses to interventions.

The first third of the paper focuses on existing predictive approaches, datasets, and evaluation strategies, with a detailed review of generative approaches (e.g. based on autoencoders, GANs, and diffusion models) and recent biological foundation models (FMs). They subdivide these models based on their function and data modality, e.g. genomic FMs for predicting transcriptomic effects of genetic variation and Protein FMs for predicting protein structure and other properties of proteins. They describe commonly-used datasets, including large-scale atlases like Tahoe-100M to synthetic data generators like Splatter. Finally, they discuss evaluation metrics focused on predictive fit, e.g. the accuracy of sequence classification or cell state classification, and discuss crucial limitations of such metrics, e.g. that they do not reflect the ability of these models to generalize to unseen perturbations.

The second third of the paper focuses on existing causal approaches, interventional data, and causal evaluation metrics. They outline different perspectives on what makes a method "causal", e.g. a mechanistic perspective that emphasizes biochemical interactions, a probabilistic perspective that emphasizes conditional independences and (conditional) invariances, and a counterfactual/interventionist perspective that emphasizes the outcomes of interventions. They review ODE-based models, causal machine learning, graphical approaches, and perturbation prediction approaches such as scCausalVI and CINEMA-OT, then review perturbation dataset such as Perturb-seq screens. They discuss various causality-relevant metrics, such as mechanistic alignment with known pathways, GRN recovery for graph-based approaches, and metrics related to the effect of perturbations.

The final third of the paper focuses on a vision for the future, including the design of mechanistic models, and alternative causal evaluation approaches based on a better use of observational data, use of biological domain knowledge from sources such as Reactome, use of experimental metadata about batch effects, and uncertainty quantification.

**Strengths:**

**Originality and significance:** The main theme of the paper is timely, given the current trends on virtual cells and foundation models in biology. While the emphasis on causal evaluation is becoming more apparent in recent works, a major novel contribution is bringing this discussion into a single paper with a quite detailed literature review.

**Clarity and quality:** The main point of the paper is quite clear and convincing and the paper follows a logical structure in its presentation. The organization of existing literature is well-done and clearly supports the position argued for by the paper.

**Weaknesses:**

## Major weaknesses

1. **Overly repetitive:** I found the paper to be quite repetitive in some sections, e.g. the subsubsections "3.1.1 Metrics" and "3.3.2 Strategies" cover a lot of the same ground - "Mechanistic alignment" in Section 3.1.1 is closely related to "Pathway fidelity tasks" in Section 3.3.2. Within Section 3.1.1, structural intervention distance (SID) is repeated both within "Intervention Validity" and "Mechanistic Alignment". I think the paper would benefit from another round or two of refining the categorization and clearly articulating the main goal of each section, and how that goal is different from the goal of the other sections.

2. **Lack of concrete contribution:** As a perspective/review paper, this work accomplishes its goal. However, looking at the ICLR call for papers, it is unclear whether such papers are meant to be in-scope for the conference (though I may have missed something). To give the paper more substance, it would be very interesting if the authors had compared existing approaches on some of the causal evaluation metrics that they discussed to reinforce their message that non-causal approaches are not sufficient for the intended purpose of virtual cells.

3. **Text-heavy:** Even as a perspective/review paper, one weakness of the paper is how dense and text-heavy it is. The literature review covers a lot of ground, and citations make up a substantial portion of the overall text. Ideally, tables and figures would be used to make the paper more readable and organized, and more equations would be used so that the paper could serve as a reference to practitioners who wish to use the causal evaluation metrics described.

## Minor weaknesses
4. **Interventions vs. counterfactuals:** This point is slightly more of personal taste, but especially in biology, I believe *most* tasks that we care about can be thought about as measuring the effects of interventions, rather than generating counterfactuals. For example, when the authors introduce the third perspective on causality at the end of page 4, they say
> "a *counterfactual* view highlights potential outcomes under interventions (e.g., asking how a tumor cell's transcriptome would change if KRAS were knocked out versus left intact)"

To me, it is best to consider this question as an *interventional* one, e.g. as a conditional treatment effect: "given what I know about the cell, what do I predict *will happen* if I perform intervention X", rather than as a *counterfactual* question, which would usually be along the lines of "what *would have happened* if I had performed X in the past?".

The distinction that I have in mind is that interventions are forward-looking, and hence have practical implications for treatment, whereas counterfactuals are backward-looking, and typically more relevant for ethical issues like assigning responsibility/blame (see [1] and responses for more background on the use of interventionist vs. counterfactual language). Throughout the paper, there seems to be a confusion between counterfactuals and interventions, e.g. on line 321, "counterfactual reconstruction error... compares predicted states against observed perturbation responses", which again I would say is interventional.

Ideally, I would prefer if the paper stuck to the interventional language except when counterfactuals are specifically needed. At the very least, it would be good for the authors to acknowledge and discuss the difference between interventions and counterfactuals: the distinction is important and it would be a disservice to propagate any confusion to a biological audience which may not be as familiar with the issues.

[1] Dawid (2020), "Decision-theoretic foundations for statistical causality"

**Questions:**

1. How do you intend to reduce the repetitiveness of the paper? (see Weakness 1)
2. Is it possible to make the paper less text-heavy and easier to parse? (see Weakness 3)
3. How do you plan to address the difference between interventions and counterfactuals? (see Weakness 4). If needed, I am happy to discuss these points further in the discussion period.

---

### Comment · Reviewer_EzGt · 2025-11-23
**Authors - please respond to reviews**

To the authors: can you please respond to these reviews? The end of the discussion period is in <10 days and it would be beneficial if the reviewers have time to engage with your responses.

---

### Note · Authors · 2025-11-24

**Comment:**

We are withdrawing the submission at this time. We appreciate the reviewers’ feedback and thank them for their effort and time.”

**Withdrawal Confirmation:**

I have read and agree with the venue's withdrawal policy on behalf of myself and my co-authors.